# Movement Disorders in Oncology: From Clinical Features to Biomarkers

**DOI:** 10.3390/biomedicines10010026

**Published:** 2021-12-23

**Authors:** Luca Marsili, Alberto Vogrig, Carlo Colosimo

**Affiliations:** 1Gardner Family Center for Parkinson’s Disease and Movement Disorders, Department of Neurology, University of Cincinnati, Cincinnati, OH 45219, USA; luca.marsili@ucmail.uc.edu; 2Clinical Neurology, Santa Maria della Misericordia University Hospital, Azienda Ospedaliero Universitaria Friuli Centrale, 33100 Udine, Italy; alberto.vogrig@gmail.com; 3Department of Neurology, Santa Maria University Hospital, 05100 Terni, Italy

**Keywords:** movement disorders, paraneoplastic, oncology, immune-checkpoint inhibitors, autoimmune

## Abstract

Background: the study of movement disorders associated with oncological diseases and anticancer treatments highlights the wide range of differential diagnoses that need to be considered. In this context, the role of immune-mediated conditions is increasingly recognized and relevant, as they represent treatable disorders. Methods: we reappraise the phenomenology, pathophysiology, diagnostic testing, and treatment of movement disorders observed in the context of brain tumors, paraneoplastic conditions, and cancer immunotherapy, such as immune-checkpoint inhibitors (ICIs). Results: movement disorders secondary to brain tumors are rare and may manifest with both hyper-/hypokinetic conditions. Paraneoplastic movement disorders are caused by antineuronal antibodies targeting intracellular or neuronal surface antigens, with variable prognosis and response to treatment. ICIs promote antitumor response by the inhibition of the immune checkpoints. They are effective treatments for several malignancies, but they may cause movement disorders through an unchecked immune response. Conclusions: movement disorders due to focal neoplastic brain lesions are rare but should not be missed. Paraneoplastic movement disorders are even rarer, and their clinical-laboratory findings require focused expertise. In addition to their desired effects in cancer treatment, ICIs can induce specific neurological adverse events, sometimes manifesting with movement disorders, which often require a case-by-case, multidisciplinary, approach.

## 1. Introduction

The study of movement disorders occurring in the context of oncological diseases highlights the wide spectrum of differential diagnoses that need to be considered, including direct (structural or compressive) etiologies and indirect (immune-mediated or treatment-related) complications. On a mechanistic standpoint, it also sheds light on the potential role of the immune system in conditions—such as paraneoplastic neurological syndromes (PNS)—which can sometimes present similarly to neurodegenerative diseases. The link between cancer and neurodegeneration is intriguing as they represent two apparently opposite phenomena: dysregulated cell proliferation and cell death, respectively, that may be linked in some specific conditions and may be influenced by genetic, environmental, and immune system-related factors [1,2,3].

Movement disorders as focal manifestations of primitive and secondary brain tumors are rare [4]. Brain tumors, particularly those affecting the basal ganglia or brainstem, may often cause variegated movement disorders. There are no robust epidemiological data, but a survey of published cases shows that hyperkinetic disorders are more common than parkinsonism (60% versus 40% of cases, respectively) [4]. Paraneoplastic movement disorders are conditions associated with antineuronal antibodies targeting intracellular or neuronal surface antigens, with different prognoses and treatment response. Immune-checkpoint inhibitors (ICIs) are drugs promoting antitumor immune response by the inhibition of the immune checkpoints [5]. They are effective treatments for several tumors, but they may cause immune-mediated movement disorders as side effects.

In the present review, we critically reappraise the phenomenology, pathophysiology, diagnostic strategies, and treatment of movement disorders associated with malignancies of the brain, with paraneoplastic conditions, and finally with the intriguing new field of cancer immunotherapy such as immune-checkpoint inhibitors (ICIs). 

## 2. Movement Disorders in the Context of Brain Malignancies

Movement disorders as focal manifestations of brain tumors are rare conditions and may manifest as hyperkinetic more than hypokinetic disorders [4]. Here, we will briefly summarize the main movement disorders found in association with brain tumors. 

*Hemichorea-hemiballism* is a rare movement disorder characterized by high amplitude, flinging movements of an entire limb (or limbs) on one side of the body, which can be violent and stressful. The acute development of hemiballismus is usually caused by focal lesions in the contralateral basal ganglia, typically (but not exclusively) in the subthalamic nucleus. Many etiologies exist for this disorder, as for example vascular causes and nonketotic hyperglycemia, the latter representing one of the most common causes [6,7]. In rare cases, tumors localized in the brain (glioma, cavernous angioma, metastases, primary central nervous system lymphoma) can be the underlying cause of this disorder [8,9,10]. Prognosis is favorable for more than 60% of the patients with complete resolution without treatment [11]. 

*Symptomatic hemidystonia* is defined as dystonia involving the ipsilateral face, arm, and leg [12]. Rare patients with symptomatic hemidystonia due to tumor, arteriovenous malformations, stroke, or hemiatrophy were described so far [12,13]. All had typical dystonic movements and/or postures, identical to those seen in idiopathic (primary) torsion dystonia. The site(s) of the lesion responsible, as defined by CT scan or pathological examination, was in the contralateral caudate nucleus, lentiform nucleus (putamen), or thalamus, or in a combination of these structures [12,13].

*Hemifacial spasm* (HFS) is characterized by involuntary unilateral contractions of the muscles innervated by the ipsilateral facial nerve, usually starting around the eyes before progressing to the lower facial muscles [14]. Its prevalence is relatively high, 9.8 per 100,000 persons [14]. The accepted pathophysiology of HFS suggests that it affects the root entry zone of the facial nerve [15]. HFS can be divided into two types: primary and secondary. Primary HFS is triggered by vascular compression whereas secondary HFS comprises all other causes of facial nerve damage. Ponto-cerebellar angle tumors are rarely associated with HFS [14]. In a series of 214 patients with HFS by Colosimo and colleagues [14], only one case (0.47%) secondary to acoustic schwannoma was found. 

*Parkinsonism* is defined as the presence of bradykinesia, in combination with at least one between resting tremor and rigidity [16]. Parkinsonism seems to be very rare in individuals with tumors located in the basal ganglia [17]. Similarly, mass lesions in the brainstem or within the posterior fossa are rarely associated with parkinsonism [17]. Conversely, supratentorial tumors sparing the basal ganglia were found to induce parkinsonism and/or rest tremor in 0.3% of the cases. Supratentorial meningioma is the most frequent cause of tumor-induced parkinsonism [17]. Bilateral thalamic tumors were also reported by a few authors to cause parkinsonism [4,17]. Basal ganglia tumors associated with parkinsonism may compress nigrostriatal neurons or their terminal axons and may induce damage to both presynaptic dopaminergic neurons and postsynaptic dopamine receptors, thus causing parkinsonian symptoms [4]. Hence, space-occupying lesions may induce similar dopaminergic nigrostriatal dysfunction as seen in idiopathic Parkinson’s disease (PD) [4]. In this regard, abnormal single photon emission computed tomography (SPECT) studies were occasionally reported in parkinsonism due to brain tumors. Benincasa and colleagues [18] reported a case of hemiparkinsonism due to frontal meningioma, which, completely resolved after tumor removal, including the SPECT-related abnormalities. Secondary hemiparkinsonism (associated with a mesencephalic tumor) may respond to dopaminergic therapies, as documented by Yoshimura and colleagues [19]. Also, an abrupt change in the response to dopaminergic therapy was described in a pre-existing parkinsonism with superimposed frontal lobe tumor [20]. Again, after tumor removal (meningioma), the patient regained a marked response to levodopa treatment [20]. Extensive infiltration of the basal ganglia and thalamus by tumors may also be associated with atypical parkinsonian syndromes, particularly progressive supranuclear palsy (PSP). The distribution of neoplastic lesions in these patients is like that of PSP-tau pathology, namely in the subthalamic nucleus and brainstem, especially the midbrain tectum and the superior cerebellar peduncle [21]. The case by Posey and Spiller (1904-05), who reported a patient with progressive ophtalmoparesis and loss of balance, regarded as the earliest reported case of PSP, had in fact a midbrain sarcoma involving the right cerebral peduncle and periaqueductal area [22]. However, parkinsonism caused by mass lesion in the brainstem is rare, and there are several possible explanations to this finding. Firstly, movement disorders may be concealed by other neurological deficits, such as signs of pyramidal dysfunction or ataxia. Secondly, signs of parkinsonism may not be present until degeneration has occurred in more than 50% of the nigrostriatal fibers. Thirdly, nigrostriatal fibers are thin, and they may be more resistant to compression than well-myelinated fibers [4].

## 3. Paraneoplastic Movement Disorders

Paraneoplastic movement disorders include, by definition, any nonmetastatic, immune-mediated, hyperkinetic, or hypokinetic conditions associated with a neoplasm [23]. The spectrum of paraneoplastic movement disorders encompasses several conditions, with acute/subacute onset, rapid evolution, and multifocal localizations [24]. PNS are conditions in which autoantibodies are produced as a reaction to antigens shared by the tumor and the nervous system [25]. This phenomenon may occur for different mechanisms. PNS-associated tumors may harbor mutations in genes encoding onconeural proteins, thus leading to the production of highly immunogenic neoantigens [26]. Alternatively, in non-paraneoplastic context, autoimmunity may be due to molecular mimicry mechanisms (e.g., similarities between foreign and self-peptides causing the cross-activation of autoreactive B-cells by pathogen-derived peptides) [26]. Antibodies directed against intracellular neuronal antigens are named onconeural antibodies and are found in association to specific cytotoxic T-cells which are supposed to have a direct pathogenic role [27]. These onconeural antibodies are strong markers for an underlying malignancy and are therefore labelled “high-risk antibodies” in the updated diagnostic criteria for paraneoplastic neurologic syndromes [28]. Despite their relevant role as biomarkers, they do not have a direct pathogenic role. Conversely, antibodies against neuronal surface antigens (NSA-Ab) have a direct pathogenic role but have an intermediate or rare association with cancer (“intermediate or low-risk antibodies”). NSA-Abs are directed against receptors, ion channels, or components of neural plasma membranes [24]. The difference between onconeural and NSA-Abs is important also for therapeutic implications, in fact, immunomodulatory treatment is mainly effective in the presence of NSA-Abs, while when autoantibodies are directed against intracellular antigens, it has less marked therapeutic impact (See also Section 5).

Following a syndromic approach, paraneoplastic movement disorders may be classified in the subsequent categories: ataxia (paraneoplastic cerebellar degeneration, now labelled “rapidly progressive cerebellar syndrome” [28], chorea, dystonia, myoclonus (and opsoclonus-myoclonus ataxia syndrome—OMS), parkinsonism, paroxysmal movement disorders, stiff person spectrum disorders, and tremor. Similar disorders can develop in autoimmune (nonparaneoplastic) encephalitides, which is broad term used to indicate inflammatory brain disorders characterized by subacute onset of working memory deficits, altered mental status or psychiatric symptoms, along with new focal CNS findings (including new-onset movement disorders or seizures), with magnetic resonance imaging (MRI) or cerebrospinal fluid (CSF) evidence of inflammatory alterations, after exclusion of alternative diagnoses [27,29]. 

Below, we will give a brief description of each of the categories in which the paraneoplastic movement disorders are classified, the main antibodies and tumors found in association [30]. The main antibodies and tumors found in association with paraneoplastic movement disorder are displayed in the Table 1.

*Ataxia* of paraneoplastic origin should be considered in cases showing a rapidly progressive cerebellar syndrome, particularly if involving predominantly the vermis over the cerebellar hemispheres [31]. Involvement of the cerebellum can present as predominant or isolated form, especially in patients with anti-Yo and anti-Tr/DNER antibodies, or in association with extra-cerebellar manifestations, which may in turn inform on the targeted antigen (e.g., subacute sensory neuropathy/limbic encephalitis are common in anti-Hu cases, jaw dystonia and parkinsonism can accompany anti-Ri syndrome, while diencephalitis/limbic encephalitis are the hallmarks of anti-Ma2 antibodies) [28,32]. 

*Chorea* of paraneoplastic origin is often accompanied by subacute cognitive decline, progressive ataxia with or without neuropathy, behavioral changes, weight loss, dysautonomic symptoms, sleep disturbances, and bulbar symptoms [24,33]. 

*Dystonia* of paraneoplastic origin is characterized by subacute onset with usually associated other symptoms like chorea, orofacial dyskinesia, stereotypies, and encephalopathy or psychiatric features [30,34]. Children and young adults are more frequently affected. 

*Myoclonus* of paraneoplastic origin has typically a subacute onset and is associated with encephalopathy, brainstem involvement, possible seizures, dysautonomia and sleep disturbances. In some cases, myoclonus can be found in association with opsoclonus and/or ataxia, thus configuring the OMS [35]. 

*Parkinsonism* of paraneoplastic origin is characterized by a subacute atypical syndrome frequently due to a brainstem encephalitis with ocular abnormalities (vertical gaze palsy) and associated sleep disorders [36,37]. PSP-like symptoms with sleep disturbances are suggestive for IgLON5-related disease, which is rarely paraneoplastic [38]. 

*Paroxysmal movement disorders* of paraneoplastic origin are characterized by sudden and repetitive dystonic or dyskinetic movements. Dystonic posturing of the face and the limbs characterizes facio-brachial dystonic seizures (LGI1-antibodies), a condition that is rarely paraneoplastic [39]. Painful dystonic posturing was described in optic neuromyelitis and associated to the AQP4 antibodies [30]. 

*Stiff person spectrum disorders* of paraneoplastic origin are a spectrum of disorders characterized by stiffness, spasms, and hyperekplexia (e.g., excessive startle reaction to sudden stimuli as noise, movement, or touch) [24]. The classic form is characterized by muscle stiffness and painful spasms, involving trunk and proximal limb muscles [24]. In the stiff-limb syndrome (SLS), stiffness is more distal and confined to a limb [30]. Some variants of this condition include the so-called progressive encephalitis with rigidity and myoclonus (PERM) [40]. 

*Tremor* of paraneoplastic origin is usually found in association with other symptoms within the context of an autoimmune encephalitis and was described with various antibodies typically associated with widespread encephalopathy [30,34]. It is important to underline that myoclonus may be misdiagnosed as tremor. 

## 4. Immune-Checkpoint Inhibitors Associated Movement Disorders

ICIs revolutionized cancer treatment, improving survivals and prognosis of several malignancies. Immune-related adverse events (irAEs) are side effects caused by ICIs. They are triggered by the inhibitions of negative regulators of the immune response with the primary aim of boosting the antitumor immunity. Therefore, ICIs may be responsible of different effects which resemble autoimmune conditions affecting several organs and systems. Indeed, endocrine, gastrointestinal, pulmonary, cardiac, renal, hematological, rheumatological and dermatological irAES were described [41]. The frequency of neurological irAEs (n-irAEs) varies from 1 to 12%, and both the CNS and peripheral nervous systems may be involved, the latter three times more [42,43]. Recently, a multi-institution group of neurologists, oncologists, and experts in irAEs developed consensus guidelines to classify the n-irAEs appropriately [42]. Seven core syndromes were defined, of which four involve the CNS and three the peripheral nervous system. IrAEs involving the CNS include immune-related (ir)-meningitis, ir-encephalitis, ir-demyelinating diseases, and ir-vasculitis [42]. The ir-encephalitis encompasses several clinical presentations of movement disorders, namely cerebellitis, OMS, and stiff person spectrum disorders/PERM [42] (Figure 1). There are clinical scenarios in which n-irAEs satisfy the criteria for the diagnosis of paraneoplastic syndromes (positive “high-risk” antibodies and compatible clinical syndrome) [44]. There is both experimental [45] and real-life experience [46] that ICIs may induce PNS. Indeed, a retrospective single-center study detected an increase (+112%) of Ma2-associated paraneoplastic syndromes diagnoses since the implementation of ICIs in France [46]. Importantly, patients with ir-encephalitis can present sometimes as movement disorders, especially in the cases linked to antiphosphodiesterase 10A-Abs [47]. 

## 5. Diagnostic Algorithm and Testing

When dealing with movement disorders with an acute/subacute onset, in general, once other secondary causes were ruled out, clinicians should consider oncological diagnoses according to the clinical presentation. Brain malignancies are easily individuated through neuroimaging studies. Differently, paraneoplastic syndromes are more difficult to diagnose, as the neurological syndrome typically antedates the discovery of a systemic cancer. Conversely, for the ICIs-related movement disorders, the history of a systemic cancer treated with immunotherapy is the most important clue. In the case of brain tumors and paraneoplastic syndromes, the goal is the diagnosis and treatment of the underlying tumor when possible. In PNS, prompt initiation of drugs which acutely modulate the immune system is necessary (e.g., steroids or intravenous immunoglobulin as first-line, followed by cyclophosphamide or rituximab in nonresponsive cases). When dealing with n-irAEs, the withdrawal of ICIs followed by steroid treatment are often needed. In both PNS and n-irAEs, however, the first step is always to exclude other (more common) diagnoses, including direct neoplastic involvement (as in the case of carcinomatous meningitis, which can be difficult to demonstrate) or neuro-infectious conditions. In case of paraneoplastic conditions, the likelihood for a tumor to be the underlying cause of immune-mediated process is higher for conditions associated with onconeural antibodies (“high-risk “antibodies) compared to that of conditions associated with NSA (“intermediate-” or “low-risk” antibodies) [24]. This occurrence is relevant in the clinical practice. In fact, for a given clinical phenotype (for example, limbic encephalitis), the antibody’s type may suggest the possibility of having associated a cancer or not, thus directing the tumor search [28]. Typically, intracellular proteins derived from tumor apoptotic cell initiate the immune response in case of PNS and, therefore, most of the high-risk antibodies are directed towards antigens located in the nucleus or cytoplasm. A flow-chart with the main antibodies to be tested, based on the clinical picture and the related probability of having associated a neoplastic condition is proposed in Figure 2.

The final diagnosis of a brain tumor is mainly based on the histologic examination of the lesion [48]. Brain tumor-derived biomarkers, and in particular wet biomarkers, constitute a growing field of interest in oncological research as an alternative for invasive tumor tissue biopsy [49]. Tumor-derived biomarkers include nucleic acids, proteins, and tumor-derived extracellular vesicles that accumulate in blood or CSF and may be routinely used in the clinical diagnostic evaluation, according to the single center’s experience [50]. Hematological malignancies involving the brain can be also easily individuated through biofluid analysis [51]. 

Paraneoplastic conditions might be diagnosed by testing the different antibodies in blood and/or CSF samples (ideally both for increasing specificity and sensitivity). The diagnosis is based on a combination of immunoblotting (used mainly for intracellular antigens), cell-based assays (CBAs) (used mainly for extracellular antigens), and tissue-based assays (TBAs) [52]. In selected cases, these tests can be supplemented by extra tests in dedicated laboratories using live cells and hippocampal rat neurons [30] or alternative tests for specific antibodies (e.g., enzyme-linked immunoassays for anti-GAD, radio-immunoassays for anti-VGCC). 

Usually, the testing strategy starts with the more common antibodies, and then if they result negative, specialized TBAs and CBAs can be used in dedicated laboratories to monitor for further fewer common antibodies [30]. However, this approach for antibody detection based on commercial kits is limited by the high rate of false positive results [53], and current diagnostic criteria recommend the use of at least two distinct techniques to confirm the test results [28].

Seronegative results are common even with widespread testing. On the opposite side, incidental findings of antibodies in patients with movement disorders not due to immune-mediated conditions may be common too and may vary according to the different laboratories and techniques adopted [30]. Possible clues that may suggest a false positive/incidental finding in antibody testing are an atypical clinical presentation and the presence of antibodies only in serum and not in CSF, or at low titers. Some antibodies like anti-LGI-1 may frequently test negative in the CSF (therefore, it is important to test serum together with CSF). Regarding the n-irAEs, the diagnosis of a definite ir-encephalitis/movement disorder associated with ICIs may be challenging and requires a clinical presentation consistent with encephalitis/movement disorder, with the exclusion of secondary causes (e.g., infectious states, cancer, and radiotherapy-induced necrosis) and the presence of CNS inflammation (documented on imaging/CSF/neurophysiological studies followed by improvement with immunomodulation—steroids—and/or ICIs discontinuation, or demonstrated directly on biopsy) [42]. 

## 6. Therapeutic Approach

Within the context of brain malignancies, the gold standard is the complete or partial tumor removal, based on its location, and/or the prompt initiation of chemo/radiotherapy [54,55]. Again, in the case of paraneoplastic conditions, the first therapeutic step is the oncological treatment of the underlying tumor (if diagnosed), and, when necessary, the administration of first-line drugs able to suppress the immune response, such as intravenous (IV) steroids, IV high-dose immunoglobulins, and plasma-exchange [24]. Second-line drugs that can be used are rituximab (monoclonal antibody anti-CD20 receptor on the B cells surface), cyclophosphamide (alkylating agent crosslinking DNA), and other drugs including azathioprine and mycophenolate mofetil. Those immunomodulatory treatments, despite depressing/modulating the immune response, are considered safe given that they do not affect the prognosis of the cancer *per se* (e.g., they do not worsen it).

Building-up a multidisciplinary team with both neurologists and oncologists is strongly encouraged in these cases. In fact, an intensive oncological follow-up and a neurological evaluation may be required if the disease tends to relapse, or if second-line treatments are required [56]. Onconeural antibodies are more frequently associated with an underlying tumor, compared to NSA antibodies. In addition, the presence of onconeural antibodies is also associated to a lesser response to immune suppressive therapies [24,34]. Unfortunately, our knowledge and guidelines on how to manage these immune-mediated conditions is based on the single-centers’ and physicians’ experience, given the scarce amount of randomized clinical trials on this topic. To the best of our knowledge, only two clinical trials was conducted on the utilization of high-dose IV immunoglobulins: in one case for stiff-person syndrome [57] and in another for LGI1/CASPR2-Ab-associated epilepsy [58], with positive results in both of them. Patients with ir-encephalitis triggered by ICIs should be managed according to National Comprehensive Cancer Network guidelines [59] ICIs should be held, and steroids administered—preferentially IV methylprednisolone— followed in case of no response or worsening by intravenous immunoglobulin or plasmapheresis. Additional treatments (including second-line immunosuppression) may be considered in refractory cases. 

Symptomatic therapy may be of relief for patients and may vary according to the specific underlying neurological symptoms. As an example, tumor-related or paraneoplastic forms of dystonia may benefit from botulinum toxin injections into affected muscles, according to standardized dosages [24]. Tumor-related parkinsonism may show a good response to levodopa [19]. Paraneoplastic chorea can be treated with dopamine depletory agents (caution is imperative with these compounds, to prevent tardive parkinsonism) [60]. Finally, muscle relaxants (benzodiazepines or baclofen) and gabapentin, may be used for stiffness [24].

## 7. Challenges and Future Directions

Although movement disorders due to focal neoplastic brain lesions are rare, this diagnosis should not be missed. Clinicians should be able to recognize or suspect the presence of a brain malignancy when dealing with a hyperkinetic or hypokinetic movement disorder with atypical features, suggestive of a secondary etiology. They should also consider that, other than tumor removal, there may be several symptomatic therapeutic options to ponder. Paraneoplastic movement disorders are even rarer, but their peculiar clinical and laboratory findings should always prompt the neurologist to search for occult tumors. In addition to their desired effects in cancer treatment, ICIs can break immune tolerance to self-antigens and induce specific n-irAEs, including movement disorders. What to do in those cases, must be discussed on a case-to-case basis. So far, our knowledge about movement disorders in oncology is limited due to their relative rarity and difficulty in promptly achieving the correct diagnosis. The introduction of ICIs in the clinical practice has opened a new window on possible adverse events induced by the immune system to consider when administering these life-saving treatments.

The effective role of antibodies found in PNS, and the related pathogenic mechanisms still needs some clarifications. Several studies pointed out the presence of these antibodies not only in paraneoplastic and immune-mediated neurological syndromes, but also in neurodegenerative diseases. In some cases, autoimmune-related neurodegeneration may be misdiagnosed as idiopathic PD, multiple system atrophy, PSP, frontotemporal dementia, or even Alzheimer’s disease [61,62,63]. Interestingly, the presence of NSA-Ab was demonstrated in these neurodegenerative diseases. The percentage of positivity to NSA-Ab antibodies among neurodegenerative diseases was estimated around 14–16% according to different studies [64,65,66]. More importantly, NSA-Ab are usually found in patients with an atypical disease progression or with atypical clinical phenotypes [63]. Further studies are needed to clarify the role of autoantibodies in patients with neurodegenerative disorders.

Another intriguing field of intersection between movement disorders and oncology is represented by the CAR T-cell therapies, which include genetically modified T-cells expressing chimeric antigen receptors (CAR T-cell). CAR T-cell treatment is an emerging strategy for hematological malignancies, and was associated with neurotoxicity [67]. Although the characterization of movement disorders caused by CAR T-cell treatments as side effects is still underreported, it would be important to further monitor its possible occurrence with future retrospective multicentric studies, given the broader application in the future of this therapeutic option [68]. The early identification of these adverse events would be essential to optimize the functional outcome and oncologic management of patients [68].

## 8. Final Remarks

In the present review, although with some limitations as the nonsystematic approach and the relatively small number of studies analyzed, we provided a comprehensive revision of movement disorders in oncological diseases, including their clinical features, associated biomarkers, and the relationship with underlying tumors. Clinicians in general, and neurologists specialized in movement disorders in particular, should be aware of these rare but serious conditions. They should be able to diagnose and then treat these conditions. Also, when dealing with a given movement disorder that could be caused by an oncological condition, clinicians should promptly start a diagnostic and therapeutic work-up based on neuroimaging and, if needed also on antibody testing. They should also consider recommending drugs that module the immune system, as steroids, or stopping current anticancer therapies if the movement disorder is a side effect of the treatment *per se* and according to its severity. 

For the next future, we envision a broader diffusion of validated kits for the detection of antibodies and other biomarkers associated with oncological conditions, a widespread knowledge of the possible CNS-associated side effects caused by ICIs (and by CAR T-cell therapy), and finally a systematic enrollment of patients in clinical trials testing putative disease-modifying drugs. Additionally, an early detection of these conditions is essential for the therapeutic success, thus increasing the patient’s chance of maintaining the greatest quality of life achievable. Also, a deep understanding of immune-mediated movement disorders will help to achieve a better knowledge of the interaction between the immune and nervous system, investigating mechanisms of cell death that are frequently found in the most common neurodegenerative disorders.

## Figures and Tables

**Figure 1 biomedicines-10-00026-f001:**
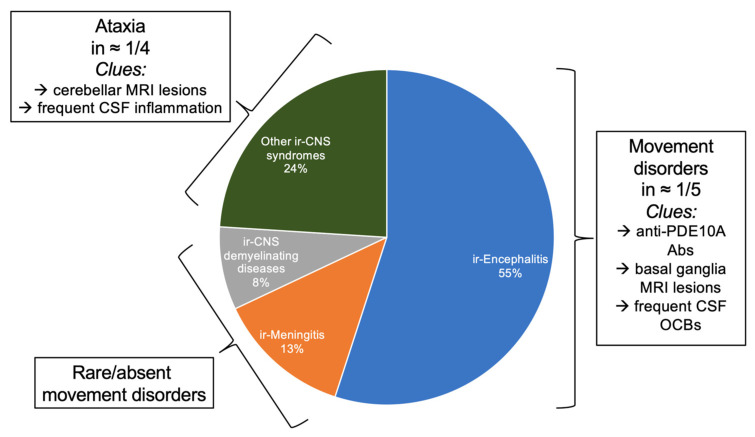
Immune-related adverse events of central nervous system associated with immune checkpoint-inhibitors. Figure is derived from observations of Marini et al. [43]. Ir-encephalitis represent 55% of cases, and movement disorders are associated in 1 out of 5 cases. Ataxia is present in 1 out of 4 cases of “Other ir-CNS syndromes”. Movement disorders are rare in other ir-related conditions. MRI, magnetic resonance imaging; CSF, cerebrospinal fluid; Anti-PDE10A Abs, antibodies anti phosphodiesterase 10A; OCBs, oligoclonal bands.

**Figure 2 biomedicines-10-00026-f002:**
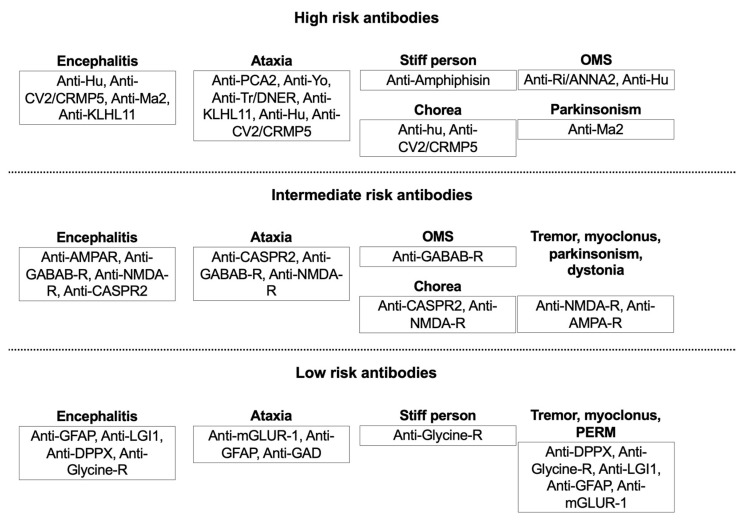
Main antibodies found in association with different paraneoplastic movement disorders, stratified per probability of having an underlying neoplastic condition. High-risk antibodies are associated with 70% probability or more of an underlying cancer. Intermediate risk antibodies with 30–70%, and low-risk antibodies with less than 30% probability of developing cancer, respectively. Each movement disorder is represented within one of three categories, together with most relevant associated antibodies. A huge overlap exists between some of these conditions, and same antibody may be found in association with different movement disorders. OMS, opsoclonus-myoclonus; OMS, opsoclonus-myoclonus ataxia syndrome; PERM, progressive encephalomyelitis with rigidity and myoclonus.

**Table 1 biomedicines-10-00026-t001:** Main antibodies and related tumors found in associations with paraneoplastic movement disorders.

Antibody	Antigen Location		Cancer Risk	Cancer Type	Associated Movement Disorders	Other Clinical Features
Anti-Yo	I		High	Breast cancer, ovary cancer	Ataxia	Uncommon
Anti-Hu/ANNA1	I		High	SCLC, NSCLC	Ataxia, chorea, OMS	SN, EM, LE
Anti-Ri/ANNA2	I		High	Breast cancer in women, lung cancer in men	Ataxia, OMS, parkinsonism, jaw dystonia, laryngospasm	BE
Anti-Tr/DNER	I		High	Hodgkin lymphoma	Ataxia	Uncommon
* Anti- KLHL11	I		High	Testicular germ cell cancer (Typically, “burned-out”)	Ataxia (including paroxysmal)	BE, myelitis, LE
Anti-PCA2	I		High	SCLC, NSCLC, breast cancer	Ataxia	Neuropathy, EM
Anti-Ma2	I		High	Testicular cancer and NSCLC	Parkinsonism	LE, BE, diencephalitis
Anti-CV2/CRMP5	I		High	SCLC and thymoma	Ataxia, chorea	SN and EM
Anti-Amphiphysin	I		High	SCLC and breast cancer	SPS	EM, SN, polyradiculoneuropathy
Anti-GABAB-R	E		Intermediate	SCLC	Ataxia, OMS	LE
Anti-CASPR2	E		Intermediate	Thymoma	Ataxia (including paroxysmal), chorea	LE, Morvan syndrome, neuromyotonia
Anti-NMDA-R	E		Intermediate	Ovarian or extra-ovarian teratoma	Dyskinesia (orofacial and limb), chorea, dystonia, stereotypies, myoclonus, ataxia, parkinsonism	Encephalitis
Anti-AMPA-R	E		Intermediate	SCLC and thymoma	Tremor	LE
Anti-LGI1	E		Low	Thymoma	Facio-brachial dystonic seizures, chorea, myoclonus, tremor	LE
Anti-GAD	S/I		Low	Rare (SCLC and thymoma)	Ataxia (including paroxysmal), SPS	LE
Anti-DPPX	E		Low	Lymphoma	Tremor, myoclonus, startle, ataxia, parkinsonism, PERM, SPS	Encephalitis
Anti-GFAP	I		Low	Ovarian teratoma and adenocarcinoma	Ataxia, tremor	Meningoencephalitis
Anti-Glycine-R	E		Low	Lymphoma, thymoma and lung cancer	PERM and SPS	LE
Anti-mGLUR-1	E		Low	Lymphoma	Ataxia, myoclonus, dystonia, tremor	Behavioral changes

I: intracellular; E: extracellular (Neural surface); LE: limbic encephalitis; BE: brainstem encephalitis; EM: encephalomyelitis; NHL: non-Hodgkin lymphoma; NSCLC: non-small cell lung cancer; OMS: opsoclonus-myoclonus ataxia syndrome; PERM: progressive encephalomyelitis with rigidity and myoclonus; S/I: synaptic intracellular; SN: sensory neuronopathy; SCLC: small cell lung cancer; SN: sensory neuronopathy; SPS: stiff-person syndrome. * Represents a newly described antibody associated to cerebellar syndrome [31].

## Data Availability

Not applicable.

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
