# Peer review of "Movement Disorders in Oncology: From Clinical Features to Biomarkers"

_biomedicines, 2021, doi:10.3390/biomedicines10010026_

Round 1
Reviewer 1 Report
This is a review of the movement disorders that occur in cancer patients, whether of tumors located in the central nervous system or in other locations (parareoplastic syndrome) or arising in the context of immunopotentiating treatments. In general, the text reads well and the information it provides is useful. Despite this, the article presents some aspects that should be improved, especially to improve the clarity of the message, before being considered suitable for publication:
- Figure 1, in the pie chart, the portion corresponding to Ataxia is yellow in color, which makes it difficult to read the text (White) that is located inside. Contrast between background and letter should be used to facilitate reading, for example by using darker letter or changing yellow for dark green.
- Table 1. Given the importance of the type of antibodies, a column should be added to the table indicating whether the antigen recognized by the antibodies is extracellular (surface) or intracellular.
- In section 3 (paraneoplastic movement disorders), it is necessary to comment that immunomodulatory treatment is mainly effective in the presence of surface antibodies, while when autoantibodies are directed against intracellular antigens, it has little therapeutic impact. This concept is addressed in section 5, however, as this section 3 is the one that is described first, a brief explanation should be made to help better describe the dichotomy of surface antigens vs intracellular antigens, something very useful for readers. clinicians who are farthest from immunology.
- At some point in the article, an explanation of the mechanisms by which these antibodies are formed (molecular mimicry, tumor neoantigens, etc.) should be provided, which is not mentioned at any time in paraneoplastic syndromes. Maybe section 3 is the right place to do it. A clinician, even with only basic training in immunology, should understand how an ovarian tumor can cause neurological disorders.
- Review abbreviations, for example line 47 mentions ICIs (immune checkpoint inhibitors) for the first time, and yet the explanation is not provided until line 54.
- In section 5 lines 239-242 you comment that in paraneoplastic syndromes it is more likely to be due to intracellular onconeuronal antibodies than surface, it would be good to discuss the statement very briefly.
- The diagnostic use of antibodies is somewhat confusing in the text (section 5). The article would benefit from an additional figure representing the diagnostic algorithm for antibodies.
- Section 6: Treatment, lines 282-288, it should be added that the treatment of paraneoplastic syndromes is immunomodulatory treatment. Likewise, it should be noted that this treatment, despite depressing / modulating the immune response, does not affect the prognosis of the cancer (it does not worsen it).
Author Response
Reviewer#1
Comment: This is a review of the movement disorders that occur in cancer patients, whether of tumors located in the central nervous system or in other locations (parareoplastic syndrome) or arising in the context of immunopotentiating treatments. In general, the text reads well and the information it provides is useful. Despite this, the article presents some aspects that should be improved, especially to improve the clarity of the message, before being considered suitable for publication:
Response: We thank the reviewer for this generous comment.
Comment 1: Figure 1, in the pie chart, the portion corresponding to Ataxia is yellow in color, which makes it difficult to read the text (White) that is located inside. Contrast between background and letter should be used to facilitate reading, for example by using darker letter or changing yellow for dark green.
Response 1: We thank the Reviewer for this observation. We have now changed the Figure 1 accordingly.
Comment 2: Table 1. Given the importance of the type of antibodies, a column should be added to the table indicating whether the antigen recognized by the antibodies is extracellular (surface) or intracellular.
Response 2: We thank the Reviewer for highlighting this important component. We have now added this information in Table 1.
Comment 3: In section 3 (paraneoplastic movement disorders), it is necessary to comment that immunomodulatory treatment is mainly effective in the presence of surface antibodies, while when autoantibodies are directed against intracellular antigens, it has little therapeutic impact. This concept is addressed in section 5, however, as this section 3 is the one that is described first, a brief explanation should be made to help better describe the dichotomy of surface antigens vs intracellular antigens, something very useful for readers. clinicians who are farthest from immunology.
Response 3: We agree with the Reviewer, and we have now added on Section 3 the following sentence: “The difference between onconeural and NSA-Abs is important also for therapeutic implications, in fact, immunomodulatory treatment is mainly effective in the presence of NSA-Abs, while when autoantibodies are directed against intracellular antigens, it has less marked therapeutic impact (See also section 5). “ (Page 3, Lines: 142-146).
Comment 4: At some point in the article, an explanation of the mechanisms by which these antibodies are formed (molecular mimicry, tumor neoantigens, etc.) should be provided, which is not mentioned at any time in paraneoplastic syndromes. Maybe section 3 is the right place to do it. A clinician, even with only basic training in immunology, should understand how an ovarian tumor can cause neurological disorders.
Response 4: We have now added a paragraph, as suggested by the Reviewer (Page:3; Lines: 126-133). “PNS are conditions in which autoantibodies are produced as a reaction to antigens shared by the tumor and the nervous system [25]. This phenomenon may occur for different mechanisms. PNS-associated tumors may harbor mutations in genes encoding onconeural proteins, thus leading to the production of highly immunogenic neoantigens [26]. Alternatively, in non-paraneoplastic context, autoimmunity may be due to molecular mimicry mechanisms (e.g., similarities between foreign and self-peptides causing the cross-activation of autoreactive B-cells by pathogen-derived peptides) [26].”
Comment 5: Review abbreviations, for example line 47 mentions ICIs (immune checkpoint inhibitors) for the first time, and yet the explanation is not provided until line 54.
Response 5: We apologize for the error. We have now spelled the term immune checkpoint inhibitors entirely on its first mention.
Comment 6: In section 5 lines 239-242 you comment that in paraneoplastic syndromes it is more likely to be due to intracellular onconeuronal antibodies than surface, it would be good to discuss the statement very briefly.
Response 6: We thank the Reviewer for this observation. We have now modified the paragraph adding the following: “This occurrence is relevant in the clinical practice. In fact, for a given clinical phenotype (for example, limbic encephalitis), the antibody’s type may suggest the possibility of having associated a cancer or not, thus directing the tumor search [28]. Typically, intracellular proteins derived from tumor apoptotic cell initiate the immune response in case of PNS and, therefore, most of the high-risk antibodies are directed towards antigens located in the nucleus or cytoplasm” (Page:7; Lines: 263-269)
Comment 7: The diagnostic use of antibodies is somewhat confusing in the text (section 5). The article would benefit from an additional figure representing the diagnostic algorithm for antibodies.
Response 7: We agree with the Reviewer, and, accordingly, we have added a new Figure (Figure 2) with a flow-chart depicting the main antibodies found in association with different paraneoplastic movement disorders, stratified per the probability of having an associated underlying neoplastic condition (namely, “high-risk,” “intermediate-risk,” and “low-risk”).
Comment 8: Section 6: Treatment, lines 282-288, it should be added that the treatment of paraneoplastic syndromes is immunomodulatory treatment. Likewise, it should be noted that this treatment, despite depressing / modulating the immune response, does not affect the prognosis of the cancer (it does not worsen it).
Response 8: We thank the Reviewer for pointing out this important aspect. We have now added a sentence to clarify this issue: “Those immunomodulatory treatments, despite depressing / modulating the immune response, are considered safe given that they do not affect the prognosis of the cancer per se (e.g., they do not worsen it).” (Page: 9; Lines: 330-332).
Reviewer 2 Report
The article is devoted to an important topic - the molecular mechanisms of movement disorders in brain tumors. Despite the fact that this is a rather rare pathology, nevertheless, the elucidation of the molecular mechanisms of its development is very important, since this can lead to the development of molecular markers of these disorders, and not only these, but other dysfunctions, and thus generally open up prospects for molecular diagnostics of paraneoplastic disorders.
The article should be recommended for publication. The only doubt is whether it corresponds to the declared theme of the special issue of the journal, because it has not been strictly proven that these disorders are based on the mechanisms of neurodegeneration.
Author Response
Reviewer#2
Comment: The article is devoted to an important topic - the molecular mechanisms of movement disorders in brain tumors. Despite the fact that this is a rather rare pathology, nevertheless, the elucidation of the molecular mechanisms of its development is very important, since this can lead to the development of molecular markers of these disorders, and not only these, but other dysfunctions, and thus generally open up prospects for molecular diagnostics of paraneoplastic disorders.
The article should be recommended for publication. The only doubt is whether it corresponds to the declared theme of the special issue of the journal, because it has not been strictly proven that these disorders are based on the mechanisms of neurodegeneration.
Response: We thank the reviewer for the general positive comment. We agree that our Review do not tackle “conventional” neurodegenerative diseases. However, the title, abstract, and outline were discussed with and approved by the Managing Editor of the Journal, purposely for this special issue. According to the Reviewer’s observation, we have now added a paragraph in section 7 “Challenges and future directions,” to underline how paraneoplastic neurological conditions are linked to neurodegenerative diseases. The paragraph reads as follows: “The effective role of antibodies found in PNS still needs some clarifications. Several studies have pointed out the presence of these antibodies not only in paraneoplastic and immune-mediated neurological syndromes, but also in neurodegenerative diseases. In some cases, autoimmune-related neurodegeneration may be misdiagnosed as idiopathic PD, multiple system atrophy, PSP, frontotemporal dementia, or even Alzheimer’s disease [61-63]. Interestingly, the presence of NSA-Ab has been demonstrated in these neurodegenerative diseases. The percentage of positivity to NSA-Ab antibodies among neurodegenerative diseases has been estimated around 14-16% according to different studies [64-66]. More importantly, NSA-Ab are usually found in patients with an atypical disease progression or with atypical clinical phenotypes [63]. Further studies are needed to clarify the role of NSA-Ab in patients with supposed neurodegenerative disorders.” (Page:10; Lines: 373-384). We have now added several new references to support our statement.
Recent lines of evidence pointed out how neurodegenerative diseases may have an autoimmune basis; hence, we believe that the study of immune-related disorders may help, at least in part, to better understanding the pathophysiology of neurodegenerative phenomena.
We hope the Reviewer may now consider the present contribution for the current special issue.